# Nickel nanoparticles set a new record of strength

A. Sharma[1], J. Hickman[2], N. Gazit [1], E. Rabkin[1] & Y. Mishin[2]

Material objects with micrometer or nanometer dimensions can exhibit much higher strength than macroscopic objects, but this strength rarely approaches the maximum theoretical strength of the material. Here, we demonstrate that faceted single-crystalline nickel (Ni) nanoparticles exhibit an ultrahigh compressive strength (up to 34 GPa) unprecedented for metallic materials. This strength matches the available estimates of Ni theoretical strength. Three factors are responsible for this record-high strength: the large Ni shear modulus, the smooth edges and corners of the nanoparticles, and the thin oxide layer on the particle surface. This finding is supported by molecular dynamics simulations that closely mimic the experimental conditions, which show that the mechanical failure of the strongest particles is triggered by homogeneous nucleation of dislocation loops inside the particle. The nucleation of a stable loop is preceded by multiple nucleation attempts accompanied by unusually large local atomic displacements caused by thermal fluctuations.

[1] Department of Materials Science and Engineering, Technion-Israel Institute of Technology, 32000 Haifa, Israel. [2] Department of Physics and Astronomy, MSN 3F3, George Mason University, Fairfax, VA 22030, USA. Correspondence and requests for materials should be addressed to E.R. (email: erabkin@technion.ac.il) or to Y.M. (email: ymishin@gmu.edu)

Mechanical properties of small crystalline objects of sub-micrometer dimensions have been the subject of intensive research during the past two decades. It is well established that deformation mechanisms change once the sample size is reduced in at least one dimension into the micrometer range[1–3]. The *Smaller is Stronger* paradigm is now universally accepted[3]. A natural question to ask is whether further down-scaling can produce objects reaching the theoretical strength of the material. In ductile metals and alloys, the latter is defined as the resolved shear stress in the primary glide plane that causes homogeneous sliding of two neighboring atomic planes past each other. An alternate definition relates the theoretical strength to the shear stress required for homogeneous, barrier-free nucleation of a dislocation loop in a perfect crystal. According to different estimates, the theoretical shear strength varies from $G/30$ to $G/8$, where $G$ is the shear modulus of the material[4]. In nanoindentation tests on well-annealed metallic single crystals, the abrupt displacement bursts are usually explained by homogeneous dislocation nucleation when the local shear stress reaches the theoretical strength[5–7]. This nucleation process is usually discussed in terms of the Hertz elastic contact theory[8], which predicts that the maximum shear stress is reached at a point some distance away from the contact. In a well-annealed single crystal, chances are high that this point is located in a defect-free region and thus a dislocation can only nucleate homogeneously[7]. This argument also applies to spherical metallic nanoparticles tested with a hard flat punch[9].

The situation is different in micro-compression or tension tests, which essentially represent a downscaled version of the classical mechanical tests[3,4,10–12]. For a cylindrical or prismatic sample, the material is homogeneously exposed to the applied stress. Thus, any dislocation or dislocation source present in the sample due to its processing history can be activated. Even in defect-free samples, dislocation half-loops or quarter-loops can nucleate at multiple locations at the surface at stresses much smaller than those required for homogeneous nucleation[7,13–16]. As a result, only a fraction of the theoretical strength can be usually achieved[7,17–19]. In faceted nanoparticles compressed by a flat punch, the facet edges at the indenter–particle interface similarly act as dislocation nucleation sites[17,18,20]. So far, a strength close to theoretical was only demonstrated for Cu[9] and Pd[21] defect-free nanowhiskers and in Au microparticles[20]. However, since the shear moduli of these metals are relatively low (e.g., 27 to 48 GPa for Au and Cu, respectively), the absolute value of their strength is only a few GPa.

Here we demonstrate that faceted single-crystalline Ni nanoparticles obtained by solid-state dewetting exhibit an ultrahigh compressive strength unprecedented for metallic materials. The three factors chiefly responsible for this strength record are the large shear modulus of Ni (78.7 GPa for {111} slip in the ⟨110⟩ direction[22]), the smooth edges and corners of the nanoparticles that reduce the stress concentration, and the thin oxide layer on the particle surface that softens the contacts with the substrate and indenter. The experimental findings reported here are supported by molecular dynamics (MD) simulations that closely reproduce the experimental conditions and provide additional insights into the dislocation mechanisms increasing the particle strength.

## Results

**Nanoparticle compression experiments.** The Ni nanoparticles were obtained by solid-state dewetting of a thin film deposited on a sapphire substrate (see Methods). X-ray diffraction scans (Supplementary Fig. 1) revealed that all particles had the same (111) orientation (in contrast to liquid-state dewetting that usually produces particles with a range of orientations, grain boundaries, and twins[23]). The diameters of the upper (111) facets of the particles ranged from 100 nm to about 1000 nm. Most of the particles had shapes close to the Wulff shape[24] and had markedly rounded edges and corners (Fig. 1a, c). Since they were formed by a long anneal at a relatively high homologous temperature ($0.77T_m$, $T_m = 1455\,°C$ being the melting temperature of Ni), they were single-crystalline and dislocation-free, which was confirmed by transmission electron microscopy (TEM) observations (Figs 1e, 2a). In addition, a thin layer of native oxide of about 4 nm in thickness was found on the surface of each particle irrespective of its size or morphology (Fig. 1f, g).

The particles were compressed in situ inside a scanning electron microscope using a Hysitron Picoindenter with a flat diamond punch of about 1 μm in diameter. To make sure that the particles were tested one at a time, each of them was examined before and after the compression, see Fig. 1c, d. Examples of experimentally measured load–displacement curves are shown in Fig. 3a for several particle sizes. The curves display a nearly linear elastic behavior up to the engineering strain of about 0.2 (see Supplementary Note 1 for explanation of the non-linear elasticity) followed by an abrupt drop in the load accompanied by a large strain burst. At this point, the particle collapses into a pancake-like disk (Fig. 1b, d and Supplementary Movie 1). Similarly, abrupt mechanical failures were previously observed in compression experiments on other defect-free structures[9,17,20]. The purely elastic behavior prior to the particle collapse was confirmed by applying a load slightly below the strain burst, unloading the particle, and then reloading it again up to the failure. No hysteresis was found during the load–unload cycle.

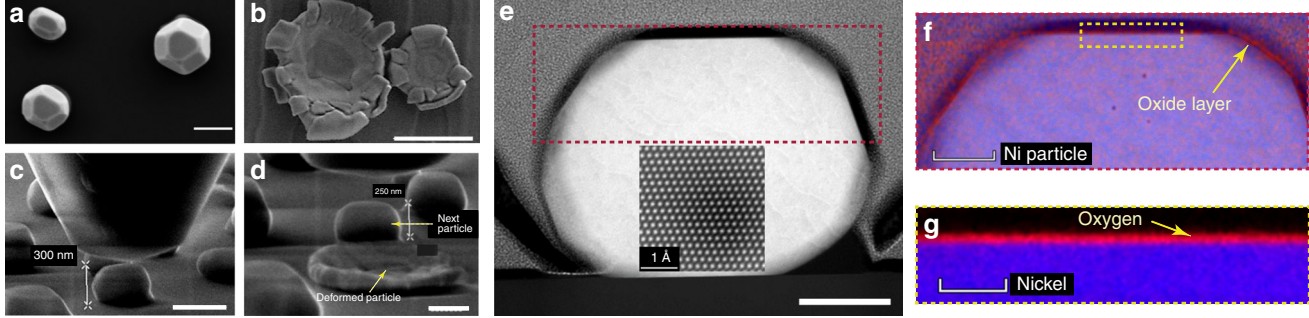

**Fig. 1** Compression tests of Ni nanoparticles. **a**, **b** High-resolution scanning electron microscopy (HR-SEM) images of particles before (**a**) and after (**b**) compression. **c**, **d** In situ images of a particle before (**c**) and after (**d**) compression. **e** STEM image of a particle prior to compression. The lattice image in the inset confirms that the particle is single crystalline. **f**, **g** STEM EDS map showing a thin, 3–4-nm oxide layer on the particle surface. The scale bars shown in **a**–**d**, **e**, **f**, and **g** correspond to 500, 100, 50, and 20 nm, respectively

The cracking at the edges of the collapsed particles can be attributed to the development of tensile hoop stresses in outer regions that were not constrained by the indenter (in most cases, the diameter of the final disk was larger than the diameter of the flat punch). TEM images in Fig. 2b, c reveal a high dislocation density and twin-like deformation structures in the collapsed particles. It was also noticed that, for the same particle size, the stress at which the particle fails correlates with the roundness of its edges and corners, with more rounded particles being stronger (Supplementary Fig. 2).

Typical engineering stress–strain curves are displayed in Fig. 3b. The engineering stress was defined as the load applied by the indenter divided by the initial area of the upper facet in contact with the punch. The engineering strain was defined as the displacement of the indenter in the loading direction divided by the initial height of the particle. It should be noted that the engineering strain could overestimate the true particle strain due to the finite machine compliance, and the deformations occurring in both the substrate and oxide layer. The compressive strength of a particle was defined as the engineering stress $\sigma$ immediately before the particle collapse. The curves in Fig. 3b display a clear size dependence of the strength with smaller particles being stronger. A similar trend was earlier found for Au particles produced by solid-state dewetting[20]. The size dependence of the strength is quantified in Fig. 3c by plotting it against the particle diameter $D$. A power law fit $\sigma = AD^{-n}$ ($A$ being a constant) gives the size exponent $n = 0.83 \pm 0.1$, which is in a good agreement

with $n$ for face-centered cubic (FCC) micro-pillars and defect-free particles (0.6–0.9)[3,20,22,25]. Note that this power law is valid over the entire range of particle sizes tested here, by contrast to the deformation behavior of spherical Fe particles[9] and defect-free Mo pillars[19].

The most remarkable finding is the ultrahigh compressive strength of the Ni nanoparticles reaching 34 GPa ($D \approx 210$ nm). This value is not only much higher than the ultimate compressive strength of bulk Ni but also exceeds the highest ultimate tensile strength of Ni nanowires fabricated by electro-deposition[26]. Furthermore, this value is almost an order of magnitude higher than the compressive strength of focused ion beam (FIB)-fabricated Ni pillars of similar diameters[25]. In fact, this compressive strength is higher than the strength of any other metal or alloy, technical ceramics or silicon nitride, and approaches the tensile strength (lower bound) of carbon nanotubes[27,28].

Converting the compressive strength $\sigma$ to the critical resolved shear stress (CRSS) $\tau$ for the $\{111\}\langle110\rangle$ slip system, the maximum CRSS measured in this work is about 10 GPa ($G/8.3$). To our knowledge, this is the highest experimentally measured shear strength for any FCC metal, including Ni itself. Note that this strength is very close to the theoretical strength of Ni. Indeed, the Frenkel model[29] predicting the maximum shear stress for uniform sliding of $\{111\}$ planes gives 15.4 and 8.7 GPa for the $\{111\}\langle110\rangle$ (full dislocations) and $\{111\}\langle112\rangle$ (partial dislocations) slips, respectively. It should be noted that Frenkel's model

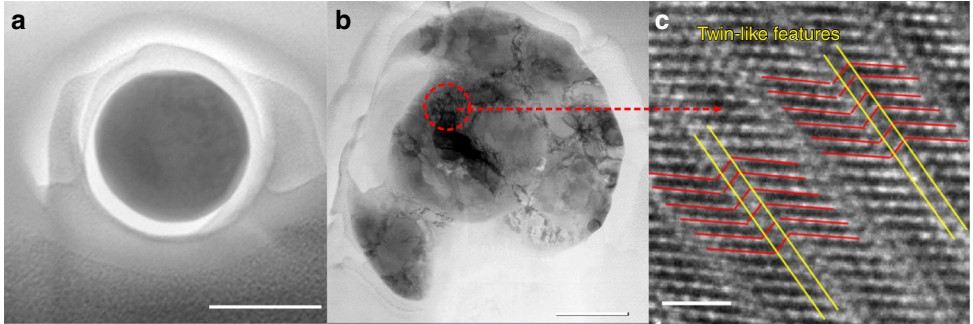

**Fig. 2** Characterization of Ni nanoparticles. **a** Plain view STEM dark field image of a pristine nanoparticle, showing the absence of dislocations or other defects. **b** STEM dark field image showing numerous defects and twin-like deformation structures after the particle failure. **c** High-resolution transmission electron microscopy (HRTEM) image from the marked area showing the twin-like structures. The scale bars shown in **a**, **b**, and **c** correspond to 200, 200, and 1 nm, respectively

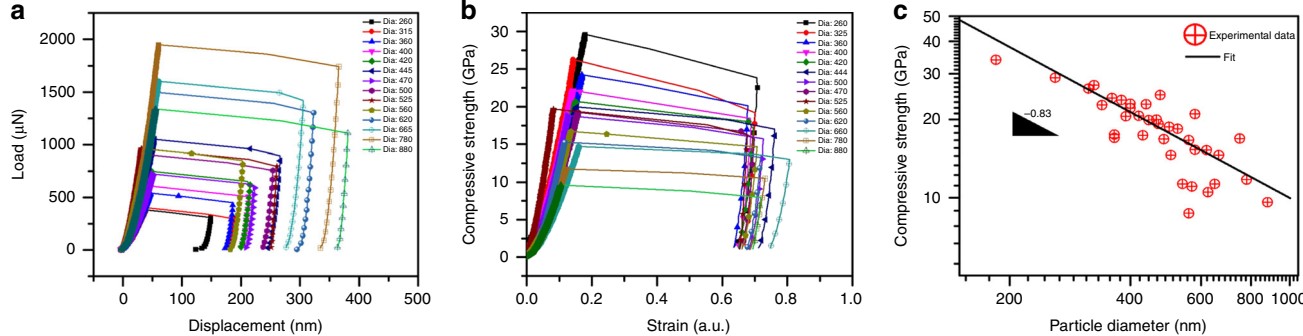

**Fig. 3** Results of mechanical testing of Ni nanoparticles. **a** Typical load–displacement curves for several particle sizes. **b** Engineering stress–strain curves demonstrate that the strength increases with decreasing particle diameter. **c** Compressive strength as a function of particle diameter

only gives an upper bound of the theoretical strength, whereas more rigorous estimates yield values in the range of 5.1–6.2 GPa ($G/16$–$G/14$).[30]

The experimental results of this work are summarized in Fig. 4 as an Ashby-type chart in which the CRSS and the particle diameter are normalized to the shear modulus $G$ and the Burgers vector $b$, respectively. For comparison, the plot includes data from the literature normalized by $G$ and $b$ for the respective slip system. The normalized CRSS measured in this work clearly exceeds the values for metallic pillars[19,25,31–36], whiskers[12,21,37], nanowires[26], equilibrated Au particles[20], and Ni₃Al nanocubes[17]. It also significantly exceeds the results of previous MD simulations of the compression of Wulff shape Ni nanoparticles with sharp corners and edges[38].

**Atomistic simulations**. To gain insights into the deformation mechanisms responsible for the ultrahigh strength of the Ni nanoparticles, atomistic simulations were performed using classical MD with an embedded-atom potential (see Methods). While previous MD simulations of nanoparticle compression were conducted at or near 0 K temperature[13–15,20,38,39], here we used a thermostat keeping the system at room temperature (300 K) to match the experimental conditions. We initially created a set of nanoparticles with different sizes but the same Wulff shape. The Wulff construction utilized the surface energies predicted by the interatomic potential. The particles initially had atomically sharp edges and corners that were not present in the experimental particles. To mimic the experimental conditions, a special algorithm was developed that started with an ideal Wulff particle and created a set of new particles with a varying degree of roundness.

The algorithm was based on gradual removal of the most ener-getic surface atoms imitating surface evaporation (see Methods). The degree of roundness of a particle was characterized by the parameter

$$\Gamma = \Gamma_0 - \sqrt{\overline{(r - \bar{r})^2}}/\bar{r}, \qquad (1)$$

where $r$ is the distance from the particle center to a given atom on the surface and the bar indicates averaging over all surface atoms. For an ideal Wulff particle $\Gamma = 0$ (no roundness), whereas a spherical particle $\Gamma$ reaches the maximum roundness value $\Gamma_0 = 0.061$.

The most typical shapes of the experimental particles appeared to be consistent with simulated particles having the roundnesses approximately within the range $0.025 < \Gamma < 0.035$. Accordingly, particles with this degree of roundness were selected for the subsequent mechanical testing. The simulated compression tests were conducted on individual particles sandwiched between two walls representing the substrate and indenter, both normal to the [111] direction of the particle. The walls were modeled by a harmonic potential (linear force) with an elastic constant $k$ and interaction distance $z_c$ (see Methods). These two parameters controlled the wall stiffness. Two choices of the wall stiffness were implemented in this work. The first choice, referred to as the hard wall, was a wall with an effective elastic modulus of 500 GPa, which is close to the stiffness of sapphire. The second choice, referred to as the soft wall, was a wall with a smaller elastic modulus tuned to 100 GPa. These two walls were chosen to represent the upper and lower bounds of possible mechanical responses of the particle surfaces covered with a relatively soft

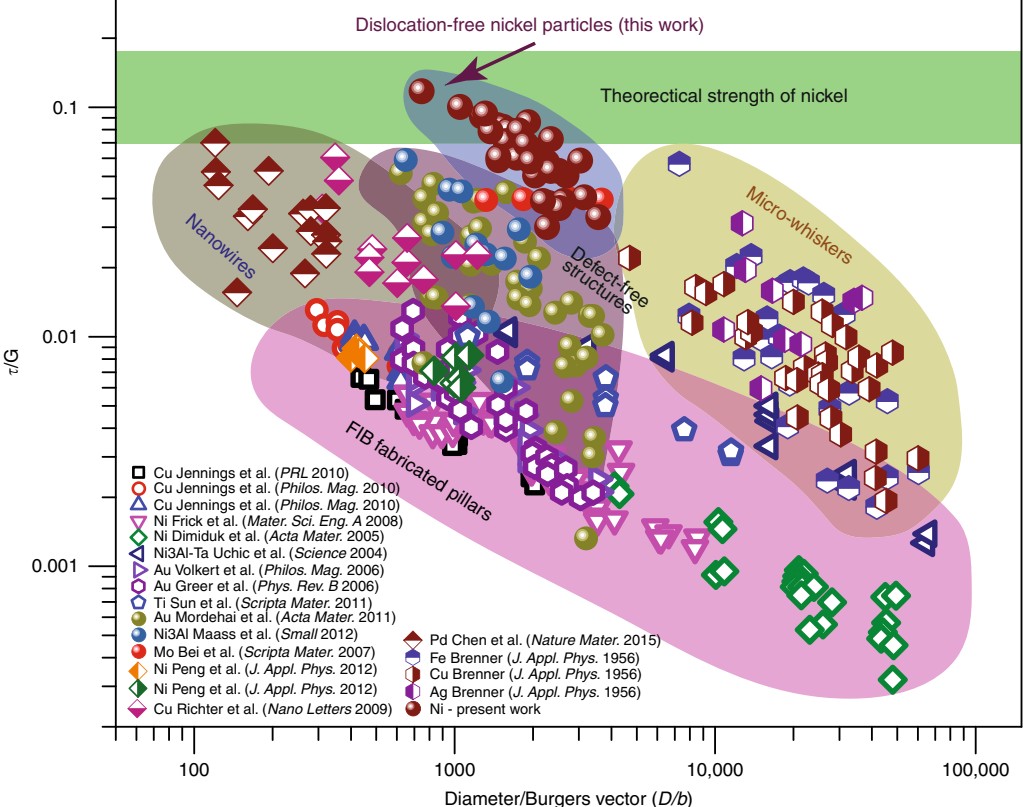

**Fig. 4** Ashby plot of CRSS $\tau$ versus diameter $D$ for Ni nanoparticles. The normalized CRSS measured in this work is compared with literature data for pillars, whiskers, nanowires, and particles of FCC and body-centered cubic metallic materials. The lower and upper bounds of the theoretical strength of Ni were obtained by first-principles calculations[30] for the {111} ⟨112⟩ slip and from the Frenkel model[29] for the {111} ⟨110⟩ slip, respectively

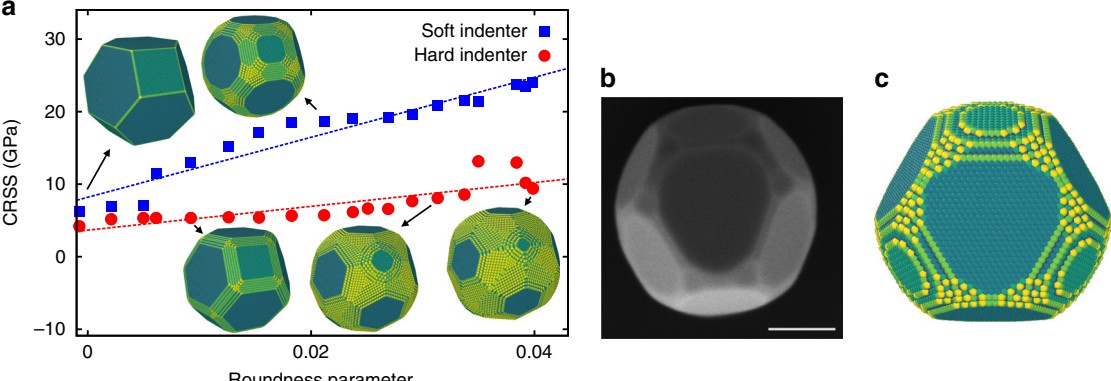

**Fig. 5** Effect of particle roundness on compressive strength in MD simulations. **a** Compressive strength of a 20 nm nanoparticle in MD simulations with the hard and soft walls as a function of the particle roundness $\Gamma$ ($\Gamma = 0$ for the ideal Wulff shape). The lines are linear fits that only serve as a guide to the eye. Typical particle shapes are shown for comparison. **b** Typical experimental particle with a rounded Wulff shape. **c** Particle from MD simulations with similar roundness. The atoms are colored according to their energy with brighter colors representing larger energy. The scale bar in (**b**) corresponds to 200 nm

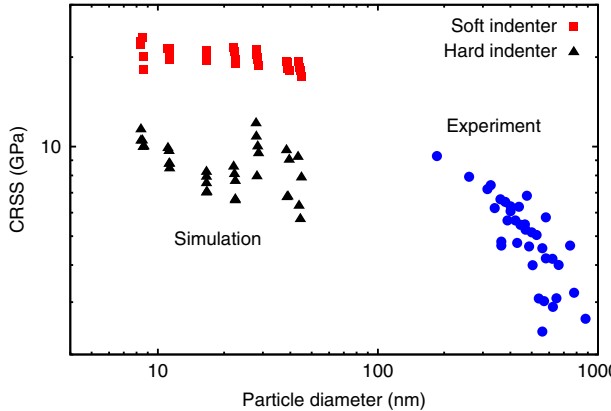

**Fig. 6** Comparison of simulated and experimental compressive strengths of Ni nanoparticles. The CRSS is plotted as a function of the particle diameter. The simulations were performed on particles with shapes similar to those in the experiments (roundness parameter $0.025 < \Gamma < 0.035$) using either hard or soft walls for particle compression

oxide layer (Fig. 1f, g). Experimentally, the elastic modulus of NiO oxide layers varies widely between 100 and 300 GPa, depending on their thickness, preparation method, and other parameters[40,41]. Each particle was thermally equilibrated at 300 K and compressed with a constant indenter speed of $1\,\text{m}\,\text{s}^{-1}$ relative to the substrate. The compressive strength was defined as the height of the first peak on the engineering stress–strain curve, which matches the experimental definition (Fig. 3).

Figure 5 demonstrates the effect of particle shape and wall stiffness on the particle strength. The particle diameter was chosen to be 20 nm, but other particle sizes gave similar results (see another example in Supplementary Fig. 3). The plot clearly shows that rounded edges and corners significantly increase the strength. A similar effect was found for cubic nanoparticles of $Ni_3Al$[42]. Importantly, softer walls strongly amplify the roundness effect. For particles with roundness similar to that in the experiments, the strength obtained with soft walls is a factor of four larger than for Wulff particles of the same size compressed by hard walls.

Results of the compression tests are summarized in Fig. 6 together with the experimental data. Note that the simulations cover smaller particle sizes than the experiment (working with larger particles was computationally prohibitive). Based on this comparison, it can be predicted that measurements on smaller particles would likely reach even higher strength levels. Specifically, it can be expected that the CRSS in smaller particles would be somewhere between the hard and soft cases and would reach about 15 GPa for 10 nm particles. Figure 6 also shows that, for smaller particle sizes, the strengthening exponent decreases as the strength levels out near the theoretical strength of the material. This can be seen more clearly in the Supplementary Fig. 4 replotting the MD and experimental results as an Ashby map in normalized coordinates.

Examination of MD snapshots has revealed that the compressive stress drops immediately after the nucleation of the first dislocation and its passage through the particle. This passage triggers surface nucleation and subsequent glide of new dislocations, their multiplication, and eventually a dislocation avalanche resulting in a large increment of strain (Supplementary Movies 2 and 3). In other words, the nanoparticle strength is totally controlled by the nucleation of the first dislocation. Figure 7 shows that the accumulation of strain is accompanied by the formation of an increasing number dislocation slip traces on the particle surface. Eventually, the particle is compressed to a pancake-like shape with fractured fringes similar to those in the experimental particles (cf. Figs. 1b, 7c).

Two mechanisms of dislocation nucleation were identified by the simulations (Fig. 8). In the particles compressed by hard walls, the first dislocation always nucleated in one of the corners of the top or bottom facet. This was preceded by multiple nucleation attempts in which a Shockley partial quarter-loop was formed and collapsed back into the corner. Eventually, the quarter-loop reached a critical size after which it began to grow, a trailing partial was nucleated from the same corner, and the full dislocation quickly propagated through the particle. A similar nucleation mechanism was observed in previous MD studies of Wulff shape particles[20,38]. The second nucleation mechanism was only found in rounded particles compressed by soft walls. In this mechanism, a partial dislocation loop nucleated *inside* the particle. Again, this was preceded by multiple nucleation attempts until a successful loop capable of further growth was formed. A trailing partial loop was then nucleated inside the leading partial loop and the full dislocation loop obtained expanded sidewise until it hit the particle surfaces. This homogeneous nucleation mechanism required a

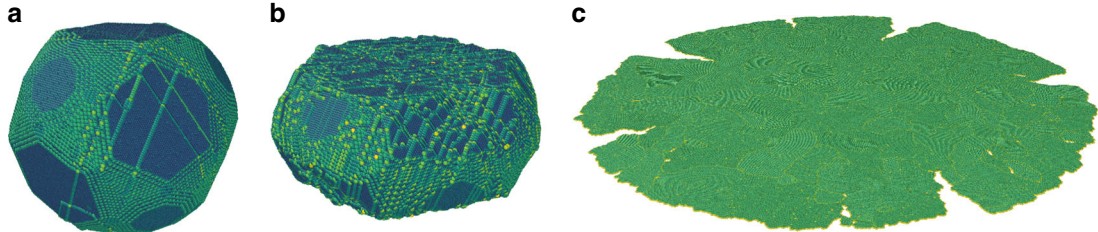

**Fig. 7** Typical Ni nanoparticle shapes at different stages of the simulated compression tests. **a** Soon after the first burst of compressive strain. Note the dislocation slip traces on the surface. **b** Later stage of compression. **c** Final shape. The initial particle diameter was 25 nm

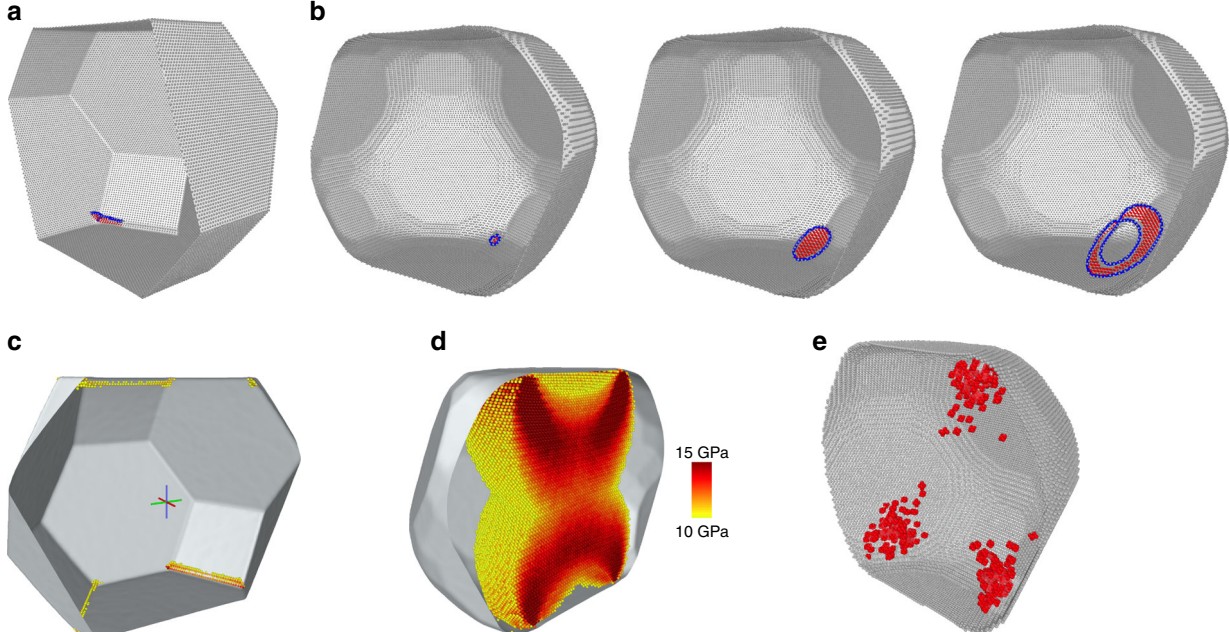

**Fig. 8** Two dislocation nucleation mechanisms in Ni nanoparticles. The initial particle diameter was 20 nm. For visual clarity, the particle was sliced in halves, perfect FCC atoms were removed, and thermal noise was eliminated by static relaxation. The Shockley partial dislocations and stacking faults are shown in blue and red, respectively. **a** Heterogeneous dislocation nucleation in the particle corner. **b** Stages of homogeneous dislocation nucleation inside the particle. **c**, **d** Resolved shear stress distribution in a Wulff particle (**c**) and in a rounded particle (**d**) just before the dislocation nucleation. Note the relatively uniform, hourglass shape, stress distribution penetrating deep into the rounded particle. The stress was averaged over 7.7 ps to remove thermal noise. Only atoms with resolved stress exceeding 10 GPa are shown, with the darker color representing larger stresses. **e** Overlap of hot spot locations (red) at several points of time showing their localization in certain zones inside the particle

higher stress level than the heterogeneous nucleation in a corner. The homogeneous nucleation mechanism was previously observed in MD simulations of single-crystalline Ni$_3$Al[43], nanoindentation simulations of metals[44,45], and most recently in compression simulations of cubic nanoparticles of Ni$_3$Al[42].

The operation of the two dislocation nucleation mechanisms was found to correlate with the stress distribution inside the particles. Figure 8c demonstrates that the compression by hard walls, especially for Wulff particles with sharp edges and corners, creates a strong stress concentration along the edges, increasing the probability of heterogeneous dislocation nucleation at those locations. By contrast, during the compression of rounded particles by soft walls, the walls could flex and bow to some extent, creating a more uniform stress distribution penetrating much deeper into inner regions of the particle (Fig. 8d). The more uniform stress distribution created a higher probability of homogeneous dislocation nucleation inside the particle. This required a higher level of the applied stress in order to reach the CRSS, making the particle stronger.

Careful examination of MD snapshots revealed that, at high stress levels, the stress components thermally fluctuated inside the particle occasionally creating localized states, called hot spots[43], with significant atomic displacements relative to the neighbors (above 17% of the equilibrium spacing). Such fluctuations favored the loop formation and gave rise to multiple nucleation attempts, until one of them turned out successful and created a dislocation loop capable of further growth. Figure 8e shows that the hot spots tend to be localized in certain zones inside the particle. Surprisingly, such zones did *not* coincide with regions of the highest resolved shear stress, hydrostatic stress, von Mises stress, or any other stress measures that we tested. In some cases, they seemed to correlate with regions of the largest resolved shear stress gradient. Tentatively, the hot zones could be linked to phonon behavior at finite temperatures, for example, to the focusing of phonons reflected from the surfaces of the smooth corners acting like parabolic mirrors. A fuller understanding of this interesting effect requires further investigations.

## Discussion

In conclusion, we have demonstrated for the first time that (111)-oriented single-crystalline Ni nanoparticles produced by solid-

state dewetting set a new record of compressive strength among metallic materials. The strength of these particles closely approaches the theoretical strength of Ni (which is about 12.5% of the shear modulus). The accompanying MD simulations have predicted about the same level of strength and revealed that the strongest particles fail by homogeneous nucleation of dislocation loops in inner regions. This work demonstrates that theoretical strength of nanoparticles can be readily achieved by eliminating all dislocations and creating optimal shapes with rounded corners and edges between crystallographic facets. Although the present work was focused on nanoparticles, single-crystalline, faceted, and dislocation-free Ni samples of more complex shapes (e.g., nanowires, nanoframes, etc.) would exhibit a similarly high strength. This work suggests a new way of producing super-strong nano- and micro-components with the aid of directed solid-state dewetting of patterned single crystalline Ni films[46]. Moreover, provided that an effective method of mass production of Ni particles similar to those studied here can be developed, such particles can find a number of technological applications such as catalytic performance, reinforcing and electrically conductive phase in polymer matrix composites, or as an additive to lubricants (ball-bearing effect). More details related to potential applications can be found in Supplementary Note 2.

## Methods

**Experimental methodology.** Thirty-nm thick nickel thin films (5 N purity) were deposited on single-side polished c-plane sapphire substrate ((0001) single crystal of a-Al$_2$O$_3$) by electron beam deposition. Prior to the deposition, the sapphire substrates (University wafers, thickness 430 μm and a miscut up to 0.2° towards the m-plane) were cleaned with acetone, ethanol, isopropanol, and deionized water, followed by a low-temperature baking at 120 °C. The deposition was performed at room temperature in a vacuum chamber with a base pressure of $5 \times 10^{-7}$ Torr at a rate of 1 Å s$^{-1}$. The as-deposited films were subject to heat treatment in a resistance quartz tube furnace under forming gas atmosphere (Ar + 10% H$_2$, 99.99% pure) at a temperature of 1050 °C for 12 h.

The morphology of the particles formed after the solid-state dewetting of the Ni film was characterized by high-resolution scanning electron microscopy (Carl Zeiss Ultra Plus) using a secondary electron detector at an acceleration voltage of 4 keV. Aberration-corrected high-resolution transmission electron microscopy (FEI Themis G2 300 80–300 keV S/TEM) was employed to analyze the microstructure in the cross-section of the particles. The cross-sectional TEM sample was prepared by the lift-out method in a dual-beam FIB (FIB; FEI Helios NanoLab DualBeam G3 UC). The final TEM sample thinning was done at an ion current of 24 pA at 2 kV to a thickness of <80 nm. The plain view TEM sample (along the loading direction) was prepared using the dual-beam FIB (FEI Helios 460 F1Lite) by employing the lift-out method with a flip stage. After mounting the sample on a Cu grid, it was thinned down from both sides with different currents and accelerating voltages. The final cleaning of the particles was performed at sample tilt of 65° with a current of 7 pA at 2 kV.

The compression tests were performed using a Hysitron PI85 picoindenter equipped with a diamond flat punch tip with a projected diameter of 1 μm . All tests were performed in a displacement-controlled mode with a constant nominal displacement rate of 1 nm s$^{-1}$. The load–displacement data were recorded after the thermal drift became <0.02 nm s$^{-1}$. The secondary electron images of the particles were collected before and after the compression to cross-check that the compression was performed on one particle at a time.

**Simulation methodology.** Interactions between Ni atoms were modeled using a well-tested embedded-atom method (EAM) potential that reproduces many experimental and first-principles data for this material[47]. Selected simulations were repeated with a different EAM potential[48] and the results revealed similar trends and led to the same conclusions. The MD and molecular statics (MS) simulations utilized the large-scale atomic/molecular massively parallel simulator (LAMMPS)[49] with the integration step of 3 fs. Smaller integration steps were also tested but gave similar results within the statistical scatter. The temperature was controlled with a Nose–Hoover thermostat with the coupling constant $T_{damp} = 0.3$(ref.[49]). The MS utilized the LAMMPS fix minimize command and the conjugate gradient minimization with $etol = 10^{-10}$ and $ftol = 10^{-10}$ (ref.[49]).

The atomic structures were visualized using the Open Visualization Tool (OVITO) software package[50].

The initial Ni nanoparticles were created by the Wulff construction using the 0 K surface energies $\gamma_{(111)} = 1.629$ J m$^{-2}$, $\gamma_{(100)} = 1.878$ J m$^{-2}$ and $\gamma_{(110)} = 2.049$ J m$^{-2}$ predicted by the EAM potential. The [11$\bar{2}$], [$\bar{1}$10], and [111] directions of the particle were aligned with the x, y, and z coordinates axes, respectively. Note that the number of atoms in the Wulff particle could only be varied by finite increments

to preserve the atomically perfect facets. The Wulff particles had atomically sharp edges and corners. The following computational procedure was developed to obtained more rounded shapes consistent with experiments. Starting with the ideal Wulff shape, atoms with the largest energy were identified and removed from the particle. By the symmetry, all atoms could be divided into groups with identical energy. The removal was applied to all atoms having the same (largest) energy. The structure was then relaxed by minimizing its total energy by MS, followed by the removal of the highest-energy atomic group from the relaxed structure. The latter was then again relaxed, followed by another removal of the highest-energy atoms, and so on. As a variant of this algorithm, two or more atomic groups (with the highest, next highest, etc., energies) were removed at a time before the relaxation. Because high-energy atoms were primarily located at edges and corners, their gradual removal tended to smooth the particle shape. This procedure could be continued for as long as needed to create a large collection of particles deviated from the original Wulff shape with a varying degree of roundness. Since this removal process targeted atoms with the highest energy, it was akin to etching or surface evaporation.

The rounding process incrementally decreased the number of atoms $N$ in the particle. Because the atoms were removed by much smaller increments than the differences between the number of atoms in ideal Wulff particles, this process created many particles with $N$ values lying between the neighboring Wulff particles. Furthermore, the rounding process could be started from any Wulff particle, thus producing large sets of rounded particles with the same (or nearly the same) number of atoms but different shapes and energies. Such particles gradually filled the gaps between the number of atoms in the Wulff particles. This is illustrated in the Supplementary Fig. 5, where the excess particle energy is plotted against the number of atoms. The excess energy is defined by $\bar{E} = (E - N\varepsilon)$, where $E$ is the total energy of the particle and $\varepsilon$ is the perfect energy of FCC Ni per atom (−4.45 eV for this EAM potential). For a given $N$, the most stable particle shape is one that minimizes $\bar{E}$. In Supplementary Fig. 5, such particles are represented by the lower envelope of the scatter plot and are closest to the equilibrium shape for any given $N$. The minimum envelope includes both ideal Wulff shapes and rounded particles in between. All points above the lower envelope represent non-equilibrium shapes and tend to be more rounded. The roundness of the particles was quantified by the parameter $\Gamma$ in Eq. (1). For computing the distance $r$ from the center of mass of the particle to its surface, the surface atoms were defined as atoms with energy 0.05 eV above $\varepsilon$.

In the simulated compression tests, both the substrate and the indenter were represented by a harmonic wall exerting a linear force. The energy associated with the wall is given by

$$\varepsilon = k(z - z_c)^2, \ z < z_c, \qquad (2)$$

where $k$ is a force constant, $z$ is the difference between the z-coordinate of an atom and the current position of the wall, and $z_c$ is the cutoff distance beyond which the atoms no longer interact with the wall. Using the harmonic wall enabled us to control the stiffness of the wall by adjusting its effective elastic modulus through the potential parameters $k$ and $z_c$. The two choices of the parameters implemented in this work were a hard wall and a soft wall with the effective elastic moduli of 500 and 100 GPa, respectively.

The position of the lower wall (substrate) was fixed while the upper wall (indenter) was moved towards the particle with a constant speed of 1 m s$^{-1}$. This process simulated the displacement-controlled conditions implemented in the experiments. The engineering strain of the particle was computed by dividing the current distance between the walls by the equilibrium distance at which they were $z_c$ away from the particle surfaces. The engineering compressive stress was obtained by $\sigma = F/A_0$, where $F$ is the total force exerted on the wall (LAMMPS output) and $A_0$ is the initial area of the top (111) surface of the particle in contact with the wall. The stress distribution inside the particle was represented by the virial stress tensor computed by LAMMPS. The simulations implemented non-periodic shrink wrapped boundary conditions in which the faces of the simulation box could change so as to always encompass the atoms in the three spatial dimensions. Before starting the compression, the particle was thermalized at 300 K by a 75 ps MD run. The compressive strength of the particle was extracted from the engineering stress–strain curve as the height of the first peak. When discussing the size dependence of the strength, we use the initial particle diameter defined as the average of the largest distances between surface atoms in the x and y directions. The CRSS was computed by multiplying $\sigma$ by the Schmid factor for the symmetrically equivalent (111) planes different from the top and bottom surfaces.

The Supplementary Fig. 6 shows that the strength of the well-equilibrated particles (lower envelope in Supplementary Fig. 5) is lower than the strength of the non-equilibrium rounded particles. Thus, part of the ultrahigh strength of the Ni particles found in the experiments and the simulations originates from their rounder than Wulff shapes. The hot spots that acted as precursors of the homogenous dislocation nucleation were identified by OVITO as regions of an other crystal structure (using the common-neighbor analysis with the default cutoff) and only existed for ~0.1 ps. Their spatial distribution shown in Fig. 8e is a superposition of 10 MD snapshots taken over a period of 1.5 ps.

## Data availability

The data that support the findings of this study are available in the Supplementary Information file or from the corresponding authors upon reasonable request. Some of the simulation data can also be downloaded in the graphical and tabulated forms from the *figshare* archive using the web links https://figshare.com/articles/Strength_data/6873683 and https://figshare.com/articles/Strength_data/6873713 with the identifiers https://doi.org/10.6084/m9.figshare.6873713.v2 and https://doi.org/10.6084/m9.figshare.6873683.v2, respectively.

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

## Acknowledgements

J.H. and Y.M. acknowledge support from the National Science Foundation, Award No. 1708314. E.R. wishes to thank the Israel Science Foundation (joint ISF-NSFC program, grant no. 2233/15) for support. The thin film deposition was performed at the Micro-Nano Fabrication and Printing Unit (MNF&PU), Technion. Collaboration between the co-authors was facilitated by the Lady Davis Research Fellowship awarded to Y.M.

## Author contributions

A.S. conducted the experiments that were designed and directed by E.R. N.G. participated in the experimental work and was instrumental in acquiring the TEM images and helping A.S. at all steps. J.H. conducted all MD simulations under Y.M.'s direction and supervision. All co-authors were engaged in discussions and contributed ideas at all stages of the work. A.S. and E.R. prepared a draft of the experimental part of the paper, while J.H. prepared a draft of the modeling part. Y.M. produced the initial draft of the complete manuscript. All co-authors participated in the manuscript editing and approved its final version.

## Additional information

**Competing interests:** The authors declare no competing interests.

