## [Peer Review File · Nature Communications]

Reviewers' comments:

Reviewer #1 (Remarks to the Author):

The manuscript „Nickel nanoparticles set a new record strength“ provides novel mechanical results on dewetted Ni particles with rounded edges supported by atomistic simulations. The main novelty of the study is that homogenous dislocation nucleation can be accomplished by micro-compression testing of rounded Ni nanoparticles. So far the sharp edges of dewetted nanoparticles would cause high stress concentrations triggering heterogeneous dislocation nucleation. As a consequence of the homogenous dislocation nucleation and the high shear modulus of Ni exceptional strength values are achieved as supported by MD simulations. In my opinion this paper should be published due to the unexpected experimental findings supported by simulation.

There are some issues which should be addressed:

The authors state that the load-displacement curves show a nearly elastic behavior, which raises the question of the origin of non-linearity in the curves. Is this a result of a non-perfect initial contact? If so why does this not cause early yielding?

The authors state in their abstract that “Ultra-strong Ni nanoparticles can have a wide range of technological application”. What do you have in mind? This point may be just for selling the story, but is otherwise not clear to me since any load bearing application would suffer from the immediate mechanical collapse of the nanoparticles upon plastic deformation due to softening into a pancake like object.

Reviewer #2 (Remarks to the Author):

In their manuscript entitled “Nickel nanoparticles set a new record of strength”, the authors presents a combined experimental-computational study on the ultimate strength of defect-free metallic nano-objects. Both with experiments and atomistic simulations, they looked at the deformation behavior of such particles under a compressive load. Experimentally, they observed for the first time in metal an ultra-high compressive strength, with a critical engineering stress of 34 GPa. The careful combined molecular dynamics/statics study evidenced the crucial role of smooth edges and soft surface oxide layers.

The presentation and the English-writting of the manuscript is crystal-clear, and the main outcomes should be understandable even by none-expert.

I appreciate the careful characterization of the experimental particle, by means of STEM and HFTEM when it was necessary. It allows the authors to conduct some of the cleanest state-of-the-art atomistic simulations that are accessible in the literature. Whereas most of the literature on nanoparticles deformation arbitrarily creates their sample, the authors developed a physics based technique to have numerical particles mimicking the experimental ones. This method, not needed to understand the paper but crucial for the accuracy and the reproducibility of the work, is well described in the supplementary materials. Similarly, the rich method section appears to facilitate reproductibility.

The authors' findings impact a large community. Material science expert with interest in mechanical properties will found this study of uppermost interest. But all scientists dealing with nanoparticles and their mechanical resilience will find in this work strong support for guiding their further work.

Besides all the aforementioned qualities, there is a key aspect that should be clarified.:

- The authors observed an ultimate compressive strength of 34 GPa. From the manuscript, it is clear that it is an engineering stress. What about the true stress? The contact area between the

indenter and the particle becoming larger, one could expect lower true stress compared to engineering stress. This is for sure not accessible experimentally, but the simulations could give some hint about it. For example, the true contact area is accessible, or even the Cauchy stress. At the very least an estimation about the variation of the stress between the "true" and the "engineering" is a must for such work.

There are other several minor aspects that could strengthen the work:

- The authors claim that "the most typical shapes of the experimental particles appeared to be consistent with simulated particles having the roundnesses [...]". How do they come to this conclusion? By visual inspection or more complex/accurate characterization techniques?
- The compression simulations are performed with a "virtual" indenter (i.e. a force purely repulsive force field). What if the author were using a "real" indenter (i.e. classical force field)? Will the friction induced by such interaction have a significant influence?
- For the sake of reproducibility, minor precision could be added into the method section:
 - What algorithm is used for the static relaxation? Which parameters (force/energy threshold)?
 - What kind of thermostat is used? Which parameters (coupling constant)?
 - The authors use static relaxation to remove thermal noise. They could also have used time averaging techniques. So, how do they ensure that the relaxation wasn't "destroying" some unstable structures, like more extended dislocations within the nanoparticle?
 - The "hot spots" are identified by their "other" crystal structure. What kind of analysis method is exactly used? Probably CNA. Which parameters?

Response to Referees' comments

We are grateful to both Referees for providing insightful comments on our paper and asking very relevant questions. This document provides our responses to the Referees and summarizes the changes made in manuscript.

Reviewer #1

Comment:

The authors state that the load-displacement curves show a nearly elastic behavior, which raises the question of the origin of non-linearity in the curves. Is this a result of a non-perfect initial contact? If so why does this not cause early yielding?

Response:

The non-linear effects mentioned by the Referee are too small to be caused by early yielding. The elastic parts of the experimental stress-strain curves are slightly non-linear for two reasons: (1) the non-linearity of the elastic properties of Ni under large strains, and (2) non-linear interactions of the nano-particle with the substrate and indenter. A new Section 2 added to the Supplementary Material presents a quantitative description of the elastic non-linearity and demonstrates a favorable comparison of the equations with typical experimental stress-strain curves (Fig. S7). A reference to this Section has been added to the main text of the paper.

Comment:

The authors state in their abstract that "Ultra-strong Ni nanoparticles can have a wide range of technological application". What do you have in mind? This point may be just for selling the story, but is otherwise not clear to me since any load bearing application would suffer from the immediate mechanical collapse of the nanoparticles upon plastic deformation due to softening into a pancake like object.

Response:

Some of the potential technological applications are mentioned in the last sentence of the paper. One of them, related to the addition of ultra-strong nanoparticles to lubricants, is discussed in more detail in the new Section added in the Supplementary Material. Furthermore, the same solid-state dewetting method as used in this paper can be applied to produce other shapes of ultra-strong single-crystalline object for MEMS and other applications. The new Section presents one example (from preliminary work of two co-authors) where a micro-spring was created by templated dewetting (Fig. S8). In most applications, the nanoparticles will not necessarily collapse; and if some of them do, the strong beneficial effect of the ultra-high strength will still remain. The new Section is referred to in a new sentence added in the end of the paper.

Reviewer #2

Comment:

The authors observed an ultimate compressive strength of 34 GPa. From the manuscript, it is clear that it is an engineering stress. What about the true stress? The contact area between the indenter and the particle becoming larger, one could expect lower true stress compared to engineering stress. This is for sure not accessible experimentally, but the simulations could give some hint about it. For example, the true contact area is accessible, or even the Cauchy stress. At the very least an estimation about the variation of the stress between the “true” and the “engineering” is a must for such work.

Response:

The instantaneous contact area was tracked during the simulations and was found to remain practically constant and equal to the initial contact area up to the first peak in the stress-strain curve. In other words, there was practically no difference between the engineering and true stresses before the first dislocation nucleation. Once a dislocation nucleated and caused dislocation multiplication and thus catastrophic collapse of the nanoparticle, the contact area rapidly increased and the two stresses diverged. However, details of the collapse process lie outside the focus of the present work are not discussed in this paper. The important point is that the ultimate strength of the Ni nanoparticles reported in the paper has not been affected by changes in the contact area.

Comment:

The authors claim that “the most typical shapes of the experimental particles appeared to be consistent with simulated particles having the roundnesses [...]”. How do they come to this conclusion? By visual inspection or more complex/accurate characterization techniques?

Response:

In this paper, the comparison was made by visual inspection. An example is shown in Figure 5 of the paper. In the future it should be possible to study the topography of the experimental particles using high-resolution AFM scans. One would then need to find a way of computing exactly the same roundness parameter as was used in the simulations. This would require a significant amount of additional work that lies beyond the scope of this paper.

Comment:

The compression simulations are performed with a “virtual” indenter (i.e. a force purely repulsive force field). What if the author were using a “real” indenter (i.e. classical force field)? Will the friction induced by such interaction have a significant influence?

Response:

We did try using a physical atomic indenter, which was Al with a scaled interatomic potential. We found this type of simulations to be computationally challenging since they involved a large number of additional atoms. Also, simulations of the actual contacts would require interatomic potentials (classical force fields) describing Ni interactions with the sapphire substrate and with the Ni oxide, and also between the Ni oxide and the diamond punch. This is totally unrealistic at the present time due to the lack of reliable interatomic potentials and many other constraints. Modeling the contacts by a generic crystal, such as Al in our attempt, would probably be possible if the computational challenges could be overcome. However, this would not represent the actual contacts much better than using the “virtual” indenter as it was done in this work.

Comment:

For the sake of reproducibility, minor precision could be added into the method section: What algorithm is used for the static relaxation? Which parameters (force/energy threshold)?

Response:

We used the LAMMPS *fix minimize* command and the conjugate gradient minimization with $etol=1e-10$ and $ftol=1e-10$ (<https://lammps.sandia.gov/doc/minimize.html>).

Comment:

What kind of thermostat is used? Which parameters (coupling constant)?

LAMMPS *fix NVT* using Nose-Hoover thermostat with $Tdamp=0.3$) (https://lammps.sandia.gov/doc/fix_nh.html)

Response:

The LAMMPS *fix NVT* using Nose-Hoover thermostat with $Tdamp=0.3$. (https://lammps.sandia.gov/doc/fix_nh.html).

Comment:

The authors use static relaxation to remove thermal noise. They could also have used time averaging technics. So, how do they ensure that the relaxation wasn't "destroying" some unstable structures, like more expended dislocations within the nanoparticle?

Response:

The averaging of atomic positions and stresses over time, mentioned by the Referee, was also carried out in this work to visualize the regions of high stress within the nanoparticle and to obtain the high stress color maps. The results from the two methods (quenching vs time average) were basically identical, suggesting that the quenching process did not result in any loss of information. For brevity, we only report the quenching results.

Comment:

The "hot spots" are identify by their "other" crystal structure. What kind of analyzes method is exactly used? Probably CNA. Which parameters?

Response:

The hot-spots were associated with atoms that were identified by OVITO as "other" type using the "conventional" CNA with the cutoff fixed at 3.00451 (which is OVITO's default for Ni). The hotspots were also also associated with regions of excessively large relative atomic displacements (~17% of the equilibrium lattice constant). The latter criterion (17%) was also applied to identify the "hot-spots" in Ref.43.

The responses to these technical questions were added to the manuscript.

We hope that these modifications have enhanced the clarity of the paper. We greatly appreciate the helpful feedback from the Referees.

REVIEWERS' COMMENTS:

Reviewer #1 (Remarks to the Author):

All questions have been satisfactorily addressed.

Reviewer #2 (Remarks to the Author):

The authors fully answered all the comments with clarity and precision. I'm convinced this work will be fully appreciated by the community!

Response to Referees' comments

We are grateful to both Referees for their supportive comments on our paper.

Reviewer #1

Comment:

All questions have been satisfactorily addressed.

Response:

Thank you.

Reviewer #2

Comment:

The authors fully answered all the comments with clarity and precision. I'm convinced this work will be fully appreciated by the community!

Response:

Thank you.